# Seizure First Aid Training For people with Epilepsy (SAFE) frequently attending emergency departments and their significant others: results of a UK multi-centre randomised controlled pilot trial

Adam J Noble [1], Dee Snape,[1] Sarah Nevitt [2], Emily A Holmes,[3] Myfanwy Morgan,[4] Catrin Tudur-Smith,[2] Dyfrig A Hughes,[3] Mark Buchanan,[5] Jane McVicar,[6] Elizabeth MacCallum,[7] Steve Goodacre [8], Leone Ridsdale,[9] Anthony G Marson[10]

For numbered affiliations see end of article.

**Correspondence to**
Dr Adam J Noble;
adam.noble@liverpool.ac.uk

## ABSTRACT

**Objective** To determine the feasibility and optimal design of a randomised controlled trial (RCT) of Seizure First Aid Training For Epilepsy (SAFE).

**Design** Pilot RCT with embedded microcosting.

**Setting** Three English hospital emergency departments (EDs).

**Participants** Patients aged ≥16 with established epilepsy reporting ≥2 ED visits in the prior 12 months and their significant others (SOs).

**Interventions** Patients (and their SOs) were randomly allocated (1:1) to SAFE plus treatment-as-usual (TAU) or TAU alone. SAFE is a 4-hour group course.

**Main outcome measures** Two criteria evaluated a definitive RCT's feasibility: (1) ≥20% of eligible patients needed to be consented into the pilot trial; (2) routine data on use of ED over the 12 months postrandomisation needed securing for ≥75%. Other measures included eligibility, ease of obtaining routine data, availability of self-report ED data and comparability, SAFE's effect and intervention cost.

**Results** Of ED attendees with a suspected seizure, 424 (10.6%) patients were eligible; 53 (12.5%) patients and 38 SOs consented. Fifty-one patients (and 37 SOs) were randomised. Routine data on ED use at 12 months were secured for 94.1% patients. Self-report ED data were available for 66.7% patients. Patients reported more visits compared with routine data. Most (76.9%) patients randomised to SAFE received it and no related serious adverse events occurred. ED use at 12 months was lower in the SAFE+TAU arm compared with TAU alone, but not significantly (rate ratio=0.62, 95% CI 0.33 to 1.17). A definitive trial would need ~674 patient participants and ~39 recruitment sites. Obtaining routine data was challenging, taking ~8.5 months.

**Conclusions** In satisfying only one predetermined 'stop/go' criterion, a definitive RCT is not feasible. The low consent rate in the pilot trial raises concerns about a definitive trial's finding's external validity and means it would be expensive to conduct. Research is required into how to optimise recruitment from the target population.

### Strengths and limitations of this study

- ► Randomisation was done remotely by computer and stratification factors and allocations concealed from those collecting baseline and follow-up data and analysing it.
- ► Participants were recruited from site's serving areas, where social deprivation was high and epilepsy control poor and so similar to those where a definitive randomised controlled trial (RCT) would likely need to focus recruitment.
- ► We completed one of the few microcostings of a self-management intervention for epilepsy.
- ► Despite recruited patient participants stating sufficient emergency department use when screened, routine data subsequently suggested ~40% did not meet the inclusion criteria.
- ► We estimated the effect of Seizure First Aid Training For Epilepsy only on the proposed primary outcome measure for a definitive RCT.

**Trial registration number** ISRCTN13871327

## BACKGROUND

International evidence shows a significant minority of people with epilepsy (PWE) frequently use hospital emergency departments (EDs).[1–4] In the United Kingdom, around 20% of PWE visit each year, of which half are admitted.[5–7] While the exact distribution of use among attendees is unclear,[8–10] ~60% may make several visits each year.[11]

Emergency hospital use by PWE has been identified as an area for potential cost savings,[12] as while expensive—approximately ~£70–90 million in England each year[8 13 14]—it

is often of little clinical benefit since most attendees have known, rather than new epilepsy and experienced uncomplicated seizures.[9 15 16] Guidance states such seizures could be managed by PWE and their significant others (SOs).[17 18] Indeed, iatrogenic harms may arise in seeking emergency care for them.

PWE visiting EDs have a characteristic profile. They report more seizures, anxiety, poorer quality of life and are more likely to live in socially deprived areas compared to the wider epilepsy population.[11 19–22] They therefore share some of the characteristics of those at increased risk of epilepsy-associated death.[23 24] In the UK, PWE visiting EDs can also be challenging to identify for research since most (~62%) are not being followed up by an epilepsy specialist,[1] primary care providers are also not always notified of ED visits by their patients[9] and because EDs do not always code the reason for an attendance in sufficient detail.[25]

While the focus on PWE attending EDs is welcomed, it needs to be translated into care improvements. Although a range of promising interventions have been suggested,[26] assessment of their utility is lacking.[27]

One intervention proposed is seizure first aid training. It has potential as PWE frequently visiting EDs and their SOs (to whom care decisions can be delegated[28 29]) express particularly low knowledge and confidence in this domain, are fearful of seizures and there are indications this leads them to seek emergency medical attention for seizures, rather than self-managing them.[30]

As no such intervention existed, we developed one—called Seizure First Aid Training For Epilepsy (SAFE). SAFE is a manualised group-based self-management course (table 1). It's rationale and development has been reported.[31] In brief, it was codesigned with health professionals, patients and carers. It was developed for delivery to groups of up to 10 patient–carer dyads by a single facilitator with knowledge of epilepsy and lasts for ~4 hours. It contains six modules centred around basic epilepsy and first aid knowledge, the recovery position, informing others about epilepsy and how to help if seizures occur, medical identification, seizure triggers and home safety. Materials include presentation slides and professionally produced videos. Its behaviour change potential has been optimised by course recipients completing a self-affirmation exercise at the start.

SAFE's efficacy now needs testing. A randomised controlled trial (RCT) would be an appropriate methodology for this. ED use over the 12 months following randomisation could be the primary outcome. It is unknown whether such a trial is feasible and what its optimal design should be.

Specifically, will patients and their SOs take part, attend SAFE and be willing and able to be followed up? It is also not possible to calculate the required sample size because the distribution of ED use is unclear. These uncertainties exist because SAFE is newly developed and no RCT has attempted to recruit the target population.

It is also unclear how to measure ED visits. Funders are encouraging trialists to use routinely recorded data to assess outcomes. In England, routine data on a person's ED use are recorded within the Hospital Episode Statistics system. This data's cost, how long it takes to obtain and its comparability to self-report are unknown.

In such circumstances, guidelines highlight the importance of pilots.[32] Their primary focus is not effect, but judging feasibility and providing information to optimise a definitive RCT's design.[33] We thus completed a pilot RCT comparing SAFE plus treatment-as-usual (TAU) or TAU alone. It had the following objectives, which will be relevant to those interested in SAFE and those researching this population: (1) estimate likely eligibility, consent and retention rates in a definitive RCT; (2) estimate annual rate of ED visits in TAU group and the likely dispersion parameter; (3) determine feasibility of measuring ED use by routine data; (4) estimate completion rates of study assessment tools; (5) estimate rates of researcher unblinding; (6) provide summary statistics to estimate effect of SAFE on ED use and its precision; (7) capture patient's and SO's views on trial participation; and (8) estimate the intervention's cost.

## METHODS
The trial's protocol has been published.[30] Here we provide a brief overview.

### Design
Parallel arm, multi-centre, external pilot RCT. Assessments with participants on the definitive trial's proposed primary and secondary outcome measures were performed at baseline prior to randomisation (T0) and 12 months later (T3, table 2). Interim assessments occurred at 3 (T1) and 6 months (T2).

SAFE was offered to the TAU alone group after T3 assessments were completed.

### Trial setting
Three hospitals in north-west England—which serve populations featuring high-social deprivation[34 35] and emergency admissions for epilepsy[21]—were recruitment sites (see Acknowledgements section). From May to December 2016 patients were invited who had attended any of these hospital's ED over the prior 12 (and with governance approval, later extended to 18) months for epilepsy (see Patient recruitment section).

### Ethical considerations and approvals
The National Research Ethics Service Committee (15/NW/0225) and Health Research Authority (166241) approved the study. An independent Trial Steering Committee monitored the trial.

### Patient recruitment
ED clinicians searched their hospitals' attendance record systems for potential participants (online supplementary file 1), screened their triage cards and posted invitations to eligible patients. Recipients had 3 weeks to return an opt-out response if not interested in participating. Those

**Table 1** SAFE course and its content

| Overview |
| --- |

SAFE was developed to be delivered to groups of ~10 patient–SO dyads by a single facilitator with epilepsy knowledge who follows a treatment manual. Materials include standardised presentation slides, videos and interactive tasks. Patients are given information packs that include copies of the slides, wallet-sized first aid instruction cards, paper epilepsy ID cards and instructions for setting up IDs on smartphone, and access to a website with the intervention's content on.

SAFE's behaviour change potential is supported by reference to Self-Affirmation Theory.

SAFE's order of presentation and content is as follows:

| Order | Topics | Learning activity | Minutes allotted |
| --- | --- | --- | --- |
| 1 | Welcome | Slide | 5 |
| 2 | Goals of this course | Slide | 2 |
| 3 | What would you like from today? | Interactive | 20 |
| 4 | True or false? | Interactive | 12 |
| 5 | Taking on information (Kindness Questionnaire) | Interactive | 10 |
| 6 | Epilepsy, seizures and how the brain works | Video | 10 |
| 7 | First aid for convulsive seizures | Interactive | 10 |
| 8 | What can you do to help someone during a seizure? | Slide | 5 |
| 9 | What not to do during a seizure | Slide | 5 |
| 10 | What to do after the seizure has stopped | Slide | 5 |
| 11 | Questions or comments? | Interactive | 10 |
| 12 | Postseizure states | Slide | 15 |
| 13 | Injuries | Slide | 2 |
| 14 | When to call an ambulance? | Slide | 10 |
| 15 | Questions or comments? | Slide | 10 |
| 16 | Refreshment break | Networking | 10 |
| 17 | Recovery position | Slide | 2 |
| 18 | Recovery position | Video | 2 |
| 19 | Let's practice the recovery position | Interactive | 15 |
| 20 | Questions or comments? | Interactive | 5 |
| 21 | Who needs to know how to help? | Interactive | 5 |
| 22 | What they need to know and why? | Slide | 5 |
| 23 | How to get this information to them. Family, friends and work colleagues: | Slide | 5 |
| 24 | How to get this information to them. Members of the public and health workers: | Slide | 5 |
| 25 | Questions or comments? | Interactive | 5 |
| 26 | Refreshment break | Networking | 5 |
| 27 | Personal stories—introduction | Slide | 2 |
| 28 | Ben's story (case vignette) | Slide | 6 |
| 29 | How to change what happened to Ben? (Carrying medical ID; triggers) | Interactive | 5 |
| 30 | Triggers | Slide | 5 |
| 31 | Knowing your triggers | Slide | 4 |
| 32 | Some ways of dealing with triggers | Slide | 4 |
| 33 | Sandra's story (case vignette) | Slide | 6 |
| 34 | How to change what happened to Sandra (Warning signs; home safety) | Interactive | 2 |
| 35 | Main points to remember, If you have epilepsy: | Slide | 3 |

Continued

| | Table 1 Continued | | |
|---|---|---|---|
| **Overview** | | | |
| 36 | Main points to remember; If you know someone with epilepsy: | Slide | 2 |
| 37 | Sources of further information | Slide | 2 |
| 38 | What's on the back table and accessing the study website* | Slide | 2 |
| 39 | Questions or comments? | Slide | 2 |
| 40 | Evaluation | | – |
| 41 | Certificates of attendance | | – |

Time in minutes: 240.
*http://www.seizurefirstaid.org.uk/Intervention/.
SAFE, Seizure First Aid Training For Epilepsy; SO, significant other.

not opting-out were telephoned by a researcher to verify eligibility and willingness to participate. Patients taking part (and their SO if they chose to take part with one) provided informed consent at an enrolment appointment with a researcher (DS). For patients, this included consent to access their routine data.

### Eligibility criteria

Table 3 details the criteria. In brief, patients were eligible if they were aged ≥16, diagnosed with epilepsy, prescribed antiepileptic drugs, could give informed consent and, when telephoned, self-reported ≥2 ED visits for epilepsy in the prior 12 months.

### Randomizsation and blinding

Patients (and their SOs) were randomised (1:1) by an online system managed by the Liverpool Clinical Trials Centre. It used a minimisation programme with a built-in random element and two stratification factors (recruitment site and whether the patient reported epilepsy stigma at baseline).

Usual care provider(s) and DS, who was responsible for recruitment and follow-up, were blinded to allocations and stratification factors. Participants could not be blinded and so were asked (and reminded at follow-ups) not to reveal their allocation to DS.

Staff (GM) organising attendance at SAFE were not involved in data collection and not blinded. The trial statistician (SN) and senior statistician (CTS) were blinded until the database was 'locked'.

### Measures to assess patient and SO participants' outcomes

Table 2 details these measures.

### Intervention

An epilepsy nurse specialist (JB) delivered SAFE within a local hospital's educational centre. A fidelity assessment found they delivered SAFE with excellent protocol adherence and competence.[36]

### Outcomes

To achieve the study objectives, rates of eligibility, consent and retention were calculated. Retention being the percentage of patients for which 12-month primary outcome data were secured. Two a priori progression criteria helped judge feasibility: (1) ≥20% of eligible patients were recruited; (2) 12-month primary outcome data were secured for ≥75% of patients.

By assessing participants on the proposed outcome measures, we obtained estimates of the distribution of ED use, measure completeness and SAFE's effect. To evaluate blinding, DS completed a 'Treatment Guess' form after each participant's T3 assessment or withdrawal. It required her to state which treatment arm she believed the participant had been randomised to. The proportion of participants willing to participate in such a trial again was ascertained and experience of adverse events calculated. Time taken to obtain routine data and the fee payable were recorded. To see if self-reported ED visits provided a reliable estimate compared with routine data, the agreement between the two data sources on how many ED visits a patient had made was explored.

### Statistical analyses

As this was a pilot RCT, a power calculation was inappropriate. Instead, the sample size was chosen to provide adequate precision to estimate the parameters required.[37]

Based on existing data,[1 14 16 38 39] it was anticipated that 12 months of attendances from each ED would allow identification of ~400 eligible patients. With a 20% consent rate, 80 patient participant accruals could be expected. With 80 patient participants we could estimate a potential drop-out rate of 25% to within a 95% CI of ±10% and a consent rate of 20% to within a 95% CI of ±4%. Assuming ED data at T3 was not available for 25% of patients, data from 60 patients would still allow robust estimation of the ED rate and dispersion parameter. Sample sizes of 24 to 50 have been recommended as 'adequate' for pilot studies.[37 40]

Analyses were documented a priori in an analysis plan and performed using SAS (V.9.4). Baseline characteristics for patient and SO participants are described using descriptive statistics and patient participants compared, on a subset, to eligible patients declining participation. Parameters are reported with 95% CIs.

**Table 2** Proposed primary and secondary outcome measures for a definitive trial used within pilot trial by assessment and participant type*

| Outcome | Domain | Participants | Measure and derivation of outcome | Baseline (T0) | 3 Months (T1) | 6 Months (T2) | 12 Months (T3) |
|---|---|---|---|---|---|---|---|
| Primary | ED visits | PWE | Routine data, Hospital Episode Statistics Accident and Emergency system.<br><br>NHS numbers of patient participants securely transferred to NHS Digital who hold the Hospital Episode Statistics system. NHS Digital then provided the number of ED visits these patients were recorded as having made in the 12 months prior to and in the 12 months following their randomisation.† | ✓ | – | – | ✓ |
| Secondary | ED visits | PWE | Self-report, 'How many times have you used Casualty/A&E over the past 12 months for epilepsy?'<br><br>(For T2 the time frame was 6 months).<br><br>Provided with seizure diary at T0 to assist. | ✓ | – | ✓ | ✓ |
| | Seizure control | PWE | Baseline: Thapar et al's[64] Seizure Frequency Scale for the prior 12 months.<br><br>T2 and T3: asked for number of epileptic seizures (of any type) since last assessment and dates of the first and most recent. Provided with seizure diary at T0 to assist. | ✓ | – | ✓ | ✓ |
| | Quality of life | PWE | Quality of Life in Epilepsy Scale-31-P (31 items)[65] | ✓ | – | ✓ | ✓ |
| | Distress | PWE; SOs | Hospital Anxiety and Depression Scale (14 items)[66]‡ | ✓ | – | – | ✓ |
| | Felt stigma | PWE; SOs | Stigma of Epilepsy Scale (9 items).[67]§ | ✓ | – | – | ✓ |
| | Burden | SOs | Zarit Caregiver Burden Inventory (22 items).[49]¶ | ✓ | – | ✓ | ✓ |
| | Adverse events | PWE | Monitored by PWE completing a checklist** | – | ✓ | ✓ | ✓ |
| | Fear of seizures | PWE, SOs | Epilepsy Knowledge and Management Questionnaire—Fears subscale (5 items).[68] | ✓ | – | – | ✓ |
| | Knowledge of what to do when faced with a seizure | PWE, SOs | Thinking About Epilepsy Questionnaire (13 items)[69] | ✓ | – | – | ✓ |
| | Confidence managing seizures/ epilepsy | PWE, SOs | PWE: Epilepsy Mastery Scale (6 items).[70]<br><br>SOs: Parents Response to Child Illness Scale-Condition Management subscale (6 items).[71] | ✓ | – | ✓ | ✓ |
| | Health economics | PWE | Client Service Receipt Inventory[72] and EQ-5D (13 items)[73] | ✓ | – | – | ✓ |
| – | Feedback on trial participation | PWE, SOs | Adapted from Magpie Trial (3 items)[74]<br><br>(1) 'If time suddenly went backward, and you had to do it all over again, would you agree to participate in the Seizure First Aid Training trial?'<br><br>(2) 'Please tell us if there was anything about the Seizure First Aid Training Trial that you think could have been done better'.<br><br>(3) 'Please tell us if there was anything about the Seizure First Aid Training Trial, or your experience of joining the trial, that you think was particularly good'. | – | – | – | ✓ |

*Unless a patient formally withdrew consent to participate in the trial, routine data on their ED use were requested and up to three attempts were made to contact patients or SO participant each time one of their follow-up assessments was due.

†To minimise cost, a single application for routine data was submitted, rather than one relating to the baseline period and one for follow-up.

‡Total anxiety and total depression scores were categorised according to the following: 0–7, 'Normal range'; 8–10, 'Suggestive of anxiety/depression'; 11–21, 'Probable anxiety/depression.'.

§Total stigma scores were categorised as follows: 0, 'No stigma'; 1–6, 'Mildly to moderately stigmatised'; 7–9, 'Highly stigmatised.

¶Total Zarit Burden scores were categorised as follows: 0–20, 'Little or no burden'; 21–40, 'Mild to moderate burden'; 41–60, 'Moderate to severe burden'; 61–88, 'Severe burden.'.

**Definitions of serious adverse event and how relatedness was assessed is presented in online supplementary file 9.

A&E, Accident & Emergency ; ED, emergency department; EQ-5D, EuroQol-5 Dimensions; NHS, National Health Service; PWE, Person with epilepsy; SOs, significant others.

**Table 3** Participant inclusion and exclusion criteria

| Participants | Inclusion criteria | Exclusion criteria |
|---|---|---|
| Patients | Established diagnosis of epilepsy (≥1 year) | Actual or suspected psychogenic non-epileptic seizures alone or in combination with epilepsy. |
| | All epilepsy syndromes and all types of focal and generalised seizures | Acute symptomatic seizures related to acute neurological illness or substance misuse. |
| | Currently being prescribed antiepileptic medication | Severe current psychiatric disorders (eg, acute psychosis) or life-threatening medical illness. |
| | Age ≥16 years (no upper age limit) | Enrolled in other epilepsy related non-pharmacological treatment studies. |
| | Visited an ED for epilepsy on ≥2 occasions within the previous 12 months (as reported by patient) | |
| | Lives within 25 miles of any of the ED recruitment sites | |
| | Able to provide informed consent, participate in intervention and independently complete questionnaires in English | |
| Significant others | A SO to the patient (eg, family member, friend) who the patient identifies as providing informal support | Severe current psychiatric disorders or life threatening medical illness. |
| | Age ≥16 years (no upper age limit) | Enrolled in other epilepsy related non-pharmacological treatment studies. |
| | Lives in the north-west area of England | |

Patient participants were permitted to take part without a significant other . Significant others could not, however, take part without a patient partner having at least consented to take part into the study.
ED, emergency department; SO, significant other.

SAFE's effect on ED use, with and without adjustment for number of ED visits prior to randomisation, was estimated using negative binomial regression (NBR) on a modified intention-to-treat basis (as defined by Del Re *et al*.[41]). Participants were included with their number of ED visits recorded with no data imputation. NBR was the prespecified statistical approach as over-dispersion (ie, large variance) was anticipated in the number of ED visits reported. Between-group differences are presented as rate ratios and, as per recommendations for hypothesis testing within pilot trials,[42] tested according to 5, 10% and 20% levels of significance.

The proportion of correct treatment guesses was determined and Cohen's Kappa computed. Bland-Altman plots compared ED visits as measured by routine data and self-report.[43]

As no consensus exists about what constitutes a clinically important reduction in ED visits,[44] average annual rate of ED visits in the SAFE+TAU and TAU alone groups postrandomisation (measured according to routine data) and the likely dispersion parameter from the adjusted NBR model were used to estimate the sample size of a definitive trial using Keene *et al*'s[45] formula. According to the formula, the number of patient participants required per group in a definitive trial to detect the size of the effect shown in the pilot study is:

$$n = \left\{ \frac{Z_{1-\beta} + Z_{1-\alpha/2}}{log(\mu_1/\mu_2)} \right\}^2 \times \left\{ \frac{\mu_1 + \mu_2}{\mu_1 \mu_2} + 2K \right\}$$

where $Z_{1-\alpha/2}$ and $Z_{1-\beta}$ are critical values of the normal distribution for specific values of alpha (α) and power (β). $\mu_1$ and $\mu_2$ are the estimated ED rates from the two treatment groups and k the negative binomial shape parameter from the associated gamma distribution which explicitly represents variability between subjects. For the calculations, alpha was set at 5%, but the dispersion parameter and power required was varied to explore differences in sample size required.

### Microcosting

Microcosting adopted the perspective of an academic non-profit making institution and was conducted in three steps[46]: (1) resource identification; (2) resource use measurement, applying the time and motion method[47]; and (3) valuation using price year 2017/2018 for local and national data. Data were analysed using Microsoft Excel 2010 to calculate the fixed and variable costs of delivering SAFE. Fixed costs included, equipment, website, freepost licence, venue hire, facilitator staff cost and facilitation resources; and assumed 11 groups/year and equipment life of 1 year. Variable costs were support staff and office costs, staff and participant

travel expenses and consumables. Total cost per group was calculated as fixed costs plus variable costs, for each group and each arm (SAFE+TAU; TAU). Mean cost per group was calculated as the sum of total costs/number of groups. Mean cost per delegate (or patient only) was calculated as sum of total cost per group in each arm/ sum of delegates (or patients only) in each arm. Results are presented as cost per training session, mean cost per delegate and mean cost per patient. The 95% central range (CR) for costs and differences were generated using Monte Carlo Simulation of 10 000 replications.

### Patient and public involvement statement

This research came about as improving education for patients and families on epilepsy is a top research priority.[48] To ensure SAFE was developed and tested in a way that met service users' needs the 'Epilepsy Society' and patient and SO representatives from the 'Brain and Spine Foundation' helped develop recruitment materials; the pilot was overseen by a Trial Steering Committee including two service user representatives; SAFE was designed by an Intervention Panel including two service user representatives and its content informed by feedback from 23 service users[31]; finally, pilot trial participants reported on the burden of participation with a view to optimising the design of a potential definitive RCT.

## RESULTS
### Participant recruitment, allocation and treatment

Of the 4016 individuals identified, 1220 (30.4%, 95% CI 29.0% to 31.8%) had visited for established epilepsy. Of these, 424 (34.8%, 95% CI 32.1% to 37.4%) were eligible; eligibility rate 10.6% (95% CI 9.6% to 11.5%, figure 1, online supplementary file 2).

Of the 424 patients invited, 53 consented; consent rate was 12.5% (95% CI 9.3% to 15.6%). Telephone contact could only be made with 203 (47.8%, 95% CI 43.1% to 52.6%) patients. The consent rate among those who could be contacted was 26.1% (95% CI 20.0% to 32.2%). The main reasons for 150 patients declining participation were 'lack of interest' (42.7%, 95% CI 34.8% to 50.6%) and being 'too busy' (22.6%, 95% CI 16.0% to 29.4%).

Of the 53 consenting patients, 51 were randomised (with 37 SOs, figure 1 gives reasons for non-randomisation). Of the 51, 26 (and 18 SOs) were allocated to SAFE+TAU and 25 (and 19 SOs) to TAU alone. Most (20, 76.9%) patients and SOs (13, 72.2%) randomised to SAFE+TAU attended a course.

### Participant demographics and epilepsy characteristics

The patient participants mean age was 39.9 years (SD 15.7, range 16–71); 29 (56.9%) were female and most (74.4%) lived in areas high in deprivation (49% in the 10% most socially deprived areas in England, table 4).

Recruited patients were comparable in age and deprivation to those declining participation. Females were over-represented (table 4).

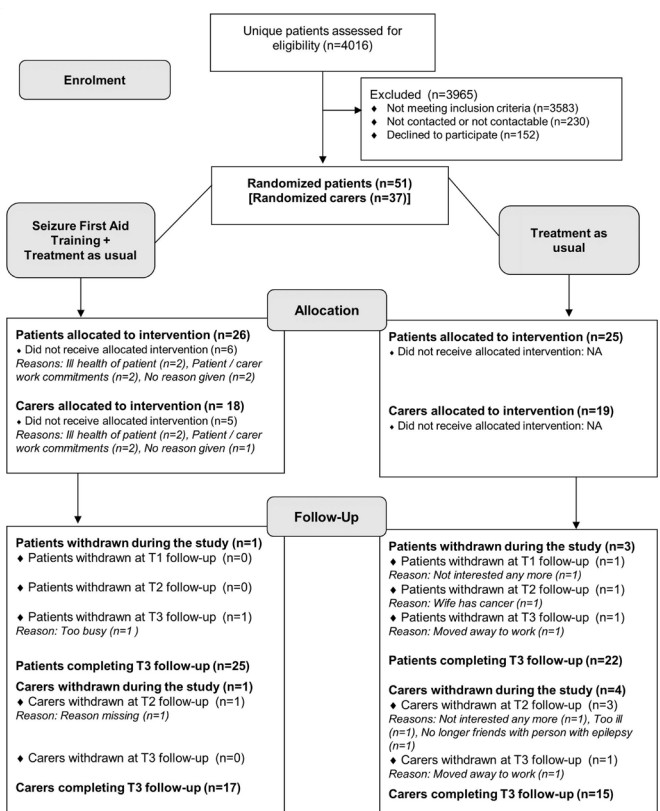

**Figure 1** CONSORT diagram of flow through the pilot trial. CONSORT, Consolidated Standards of Reporting Trials; SOs, significant others.

Recruited patients had an epilepsy diagnosis for a median of 17.3 years. Most (62.8%) patients reported ≥10 seizures in the previous year and having seen a neurologist (74.5%, table 4). Participants' mean Quality of Life-31-P score was low at 48.3 (SD 17.3; range 17.1 to 79.5). Twenty-six (50.9%) patients had 'probable' clinical anxiety. Most (n=42, 82.3%) patients reported feeling stigmatised by epilepsy; 15 (29.4%) highly. The treatment groups were broadly similar in demographics and epilepsy characteristics.

SO participants were typically a partner/spouse (43.2%) and most (89.2%) had daily patient contact (online supplementary file 3). With a mean Zarit Caregiver burden score of 18.9 (SD=12.51), SOs typically reported 'little or no burden'[49] but anxiety was high[50]; 15 (40.5%) SOs had 'probable' clinical anxiety.

### Participant retention
#### Proposed primary outcome measure

Of the randomised patients, 3 (5.8%) formally withdrew over follow-up, removing access to their routine data. Primary outcome data on ED use at 12 months (and for the 12 months prior to randomisation) were available for the remaining 48 patients, giving a retention rate of 94.1%.

#### Proposed secondary outcome measures

Thirty-seven (72.5%) randomised patients and 21 (56.8%) SOs attended their 12 months questionnaire

**Table 4** Characteristics of eligible patients who did and did not agree to participate

| Variable | | Agreed to participate | | | Did not agree to participate |
|---|---|---|---|---|---|
| | | **SAFE+TAU** | **TAU** | **Combined** | |
| | | **(n=26)** | **(n=25)** | **(n=51)** | **(n=379)\*** |
| Sex: n (%) | | | | | |
| Male | | 10 (38.5%) | 12 (48.0%) | 22 (43.1%) | 192 (50.7%) |
| Female | | 16 (61.5%) | 13 (52.0%) | 29 (56.9%) | 187 (49.3%) |
| Age at presentation to ED (years) | | | | | |
| Mean | | 39.2 | 40.7 | 39.9 | 40.6 |
| SD | | 13.96 | 17.52 | 15.66 | 16.83 |
| Median (min, max) | | 37.1 (18.9 to 69.9) | 41.4 (16.5 to 71.3) | 38.8 (16.4 to 71.3) | 37.5 (16.2 to 84.4) |
| IMD† | | | | | |
| IMD decile: 1 | n (%) | 11 (42.3%) | 14 (56.0%) | 25 (49.0%) | 188 (49.6%) |
| IMD rank | Median (min, max) | 574.0 (44 to 3202) | 1231.5 (48 to 2166) | 673.0 (44 to 3202) | 924.5 (28 to 3107) |
| IMD decile: 2–3 | n (%) | 7 (26.9%) | 6 (24.0) | 13 (25.4%) | 82 (21.6%) |
| IMD rank | Median (min, max) | 6785 (4649 to 8281) | 6665 (3989 to 7816) | 6785 (3989 to 8281) | 5309 (3291 to 9835) |
| IMD decile: 4–6 | n (%) | 2 (7.7%) | 4 (16.0%) | 6 (11.8%) | 59 (15.6%) |
| IMD rank | Median (min, max) | 128 366 (9881 to 15791) | 119 244 (11 480 to 16004) | 119 244 (9881 to 16004) | 144 599 (10 277 to 19 659) |
| IMD decile: 7–10 | n (%) | 5 (19.2%) | 1 (4.0%) | 6 (11.8%) | 50 (13.2%) |
| IMD rank | Median (min, max) | 276 422 (24 971 to 31 002) | 327 244 (32 724 to 32 724) | 288 766 (24 971 to 32 724) | 226 733 (19 826 to 32 396) |
| | Missing | 1 (3.8%) | 0 (0.0%) | 1 (2.0%) | 0 (0.0%) |
| Ethnicity: n (%) | | | | | |
| White | | 25 (96.2%) | 23 (92.0%) | 48 (94.1%) | – |
| Asian/Asian British | | 0 (0.0%) | 0 (0.0%) | 0 (0.0%) | – |
| Black/African/Caribbean/Black British | | 1 (3.8%) | 1 (4.0%) | 2 (3.9%) | – |
| Mixed/multiple ethnic groups | | 0 (0.0%) | 0 (0.0%) | 0 (0.0%) | – |
| Other ethnic group | | 0 (0.0%) | 1 (4.0%) | 1 (2.0%) | – |
| Other significant medical history: n (%) | | | | | |
| No, none | | 13 (50.0%) | 10 (40.0%) | 23 (45.1%) | – |
| Yes, another medical condition/s | | 10 (38.5%) | 13 (52.0%) | 23 (45.1%) | – |
| Yes, a psychiatric condition | | 1 (3.8%) | 0 (0.0%) | 1 (2.0%) | – |
| Yes, both medical and psychiatric conditions | | 2 (7.7%) | 2 (8.0%) | 4 (7.8%) | – |
| Education: n (%) | | | | | |
| O' levels/GCSEs/level 1 or 2 NVQ | | 13 (50.0%) | 14 (56.0%) | 27 (53.0%) | – |
| A' levels/level 3 NVQ | | 5 (19.2%) | 3 (12.0%) | 8 (15.7%) | – |
| University degree/graduate certificate or diploma | | 5 (19.2%) | 5 (20.0%) | 10 (19.6%) | – |
| Postgraduate university degree (eg, PGCE, MSc, MA, PhD) | | 2 (7.7%) | 0 (0.0%) | 2 (3.9%) | – |
| Missing | | 1 (3.9%) | 3 (12.0%) | 4 (7.8%) | – |
| Time since epilepsy diagnosis (years) | | | | | |
| Mean | | 19.9 | 22.6 | 21.2 | – |
| SD | | 14.85 | 18.38 | 16.57 | – |
| Median (mix, max) | | 16.8 (1.8 to 53.9) | 19.3 (1.7 to 64.9) | 17.3 (1.7 to 64.9) | – |
| Time since last epileptic seizure (days) | | | | | |
| Mean | | 53.6 | 40.1 | 47 | – |
| SD | | 101.1 | 61.21 | 83.34 | – |
| Median (min, max) | | 14.0 (1 to 340) | 10.5 (0 to 235) | 14.0 (0 to 340) | – |

Continued

**Table 4** Continued

| Variable | Agreed to participate | | | Did not agree to participate |
|---|---|---|---|---|
| | SAFE+TAU | TAU | Combined | |
| | (n=26) | (n=25) | (n=51) | (n=379)* |
| Missing | 3 (11.5%) | 3 (12.0%) | 6 (11.8%) | – |
| Number of epileptic seizures in the last 12 months | | | | |
| 0–3 | 4 (15.3%) | 3 (12.0%) | 7 (13.8%) | – |
| 6-April | 1 (3.8%) | 5 (20.0%) | 6 (11.7%) | – |
| 9-July | 3 (11.5%) | 3 (12.0%) | 6 (11.7%) | – |
| 10 or more | 18 (69.2%) | 14 (56.0%) | 32 (62.8%) | – |

*Includes seven patients who could not be contacted due to an incorrect postal address and two patients who were not sent an invite letter in error. min=minimum, max=maximum.
†The IMD ranks every small area in England from 1 (most deprived area) to 32 844 (least deprived area). IMD rank and decile missing for one recruited participant as they resided in Wales.
GCSE, General Certificate of Secondary Education; IMD, Index of Multiple Deprivation; NVQ, National Vocational Qualification; SAFE, Seizure First Aid Training For Epilepsy; TAU, treatment-as-usual.

appointment (T3). The extent to which measures were completed at these and the interim appointments varied (online supplementary file 4). Self-report data on ED use at T3 was obtained from only 34 patients, giving a retention rate on this measure of 66.7% patients.

### ED use
#### Baseline, prior to randomisation
Routine data for the 48 patients for whom consent was maintained showed they made 122 ED visits in the 12 months before randomisation (online supplementary file 5). The mean was 2.5 and median 2 (range 0 to 12). ED use was slightly higher for TAU participants (table 5).

Despite only consenting patients who when telephoned reported ≥2 ED visits in the prior 12 months, routine data indicated 4 (8.3%) had not made any visits during the prior 12 months. A further 19 (39.6%) made only one.

#### At 12 months, effect of SAFE
Compared with the 12 months prior to randomisation, mean ED use over follow-up reduced for the SAFE+TAU group from 2.1 visits to 1.8 (difference −0.3). For the TAU group, it increased from 3 visits to 3.4 (difference 0.4, table 5). Unadjusted NBR estimated the visit rate was 46% lower in the SAFE+TAU group compared with the TAU group (rate ratio=0.54; Vuong's test $z=−0.17$,

**Table 5** Number of ED visits patient participants made according to routine data

| Number of ED visits | SAFE+TAU (n=26) | TAU (n=25) | Total (n=51) |
|---|---|---|---|
| In the 12 months to baseline | | | |
| Mean | 2.1 | 3 | 2.5 |
| SD | 2.22 | 2.76 | 2.51 |
| Median (min, max) | 1 (0 to 10) | 2 (1 to 12) | 2 (0 to 12) |
| Missing | 1 (3.8%) | 2 (8.0%) | 3 (5.9%) |
| In the 12 months following randomisation | | | |
| Mean | 1.8 | 3.4 | 2.6 |
| SD | 3.14 | 4.78 | 4.05 |
| Median (min, max) | 1 (0 to 12) | 2 (0 to 20) | 1 (0 to 20) |
| Missing | 1 (3.8%) | 2 (8.0%) | 3 (5.9%) |
| Change from baseline over 12 months following randomisation | | | |
| Mean | −0.3 | 0.4 | 0.1 |
| SD | 1.99 | 3.81 | 2.99 |
| Median (min, max) | 0 (-4 to 5) | 0 (−5 to 16) | 0 (−5 to 16) |
| Missing | 1 (3.8%) | 2 (8.0%) | 3 (5.9%) |

ED, emergency department; SAFE, Seizure First Aid Training For Epilepsy; TAU, treatment-as-usual.

**Table 6** Between-group intervention differences in number of ED visits

| Model and parameter | Parameter | CI 95% | 90% | 80% | P value |
|---|---|---|---|---|---|
| 12 Months following randomisation according to routine data* | | | | | |
| Negative binomial: SAFE+TAU (rate ratio) | 0.54 | 0.24 to 1.18 | 0.28 to 1.04 | 0.32 to 0.90 | 0.12 |
| Negative binomial: dispersion parameter | 1.53 | 0.67 to 2.39 | 0.80 to 2.25 | 0.96 to 2.09 | NA |
| Vuong's test† | −0.17 | NA | NA | NA | 0.87 |
| 12 Months following randomisation according to routine data with adjustment for baseline ED visits | | | | | |
| Negative binomial: SAFE+TAU (rate ratio) | 0.62 | 0.33 to 1.17 | 0.36 to 1.06 | 0.41 to 0.94 | 0.14 |
| Negative binomial: Baseline ED visits (rate ratio) | 1.33 | 1.18 to 1.52 | 1.20 to 1.49 | 1.23 to 1.45 | <0.001 |
| Negative binomial: Dispersion parameter | 0.69 | 0.17 to 1.21 | 0.26 to 1.13 | 0.35 to 1.03 | NA |
| Vuong's test† | −0.13 | NA | NA | NA | 0.90 |

*Analysis based on 48 patient participants.

†Vuong's test p value interpretation, a significantly negative parameter value favours the negative binomial model and significantly positive favours the zero-inflated negative binomial model. A non-significant value indicates no significant difference between the models therefore the simpler negative binomial model is preferred.[75]

.ED, emergency department; SAFE, Seizure First Aid Training For Epilepsy; TAU, treatment-as-usual.

p=0.87). In the adjusted model, SAFE's effect reduced (rate ratio=0.62). Between-group differences were not significant at the 5% or 10% alpha level in either the unadjusted or adjusted model. The effect was significant in both at the 20% level (table 6). The dispersion parameter under the adjusted NBR model was k=0.69 (CI 0.17 to 1.21).

### Obtaining routine data and its correspondence with self-report

Routine data took 8.5 months to secure, arriving ~9 months after T3 assessments finished. The direct cost was £6960. Online supplementary file 6 shows substantial work on behalf of the research team to obtain the data was required. In one instance, an appeal against a decision to reject the application—5 months in—was necessary.

Routine data did not match patient self-report in most cases. Forty-two patients had self-report and routine data on ED use at baseline. Only 3 (7.1%) reported the same number of visits as their routine data indicated. Most (76.2%) patients reported more (by 3.8 visits on average, figure 2A). There was greater agreement between routine and self-report data at T3; 11 (32.4%) patients reported the same number of visits as their routine data (figure 2B).

### Blinding

The researcher correctly guessed which of the two treatment arms 35 patient participants had been allocated to by the randomisation process; unblinding rate 68.6% (CI 54.1% to 80.9%). The chance-corrected kappa statistic of 0.37 (CI 0.12 to 0.63) equated to 'fair' agreement.

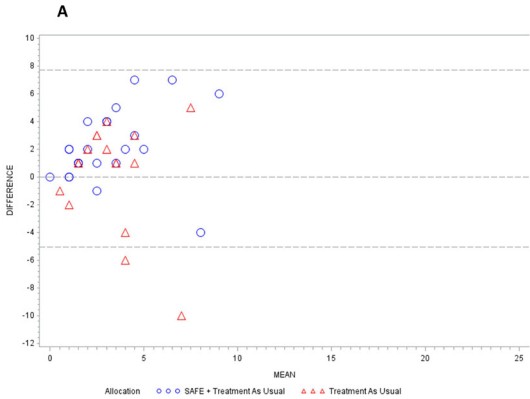

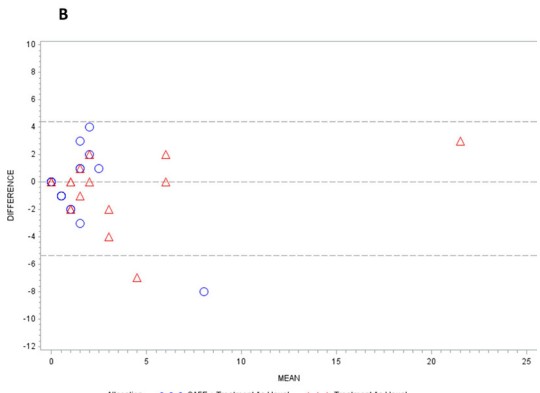

**Figure 2** (A) Bland-Altman plot of agreement between self-reported ED use and routine data on ED use in 12 months prior to randomisation. (B) Bland-Altman plot of agreement between self-reported ED visits and routine data on ED use over 12 months following randomisation. ED, emergency department; SAFE, Seizure First Aid Training For Epilepsy.

**Table 7** Required sample size for a definitive trial to detect estimated effect of SAFE+TAU on ED use

| Dispersion parameter (k) | ~47% Reduction (from 3.4 to 1.8 visits) in 12 months compared with TAU | |
| --- | --- | --- |
| | N per group (α=0.05; 80% power, β=0.2) | N per group (α=0.05; 90% power, β=0.1) |
| 0.17 | 123 | 164 |
| 0.5 | 191 | 255 |
| 0.69 | 230 | 308 |
| 1 | 293 | 393 |
| 1.21 | 337 | 451 |

Dispersion parameter taken from the adjusted NBR model in the pilot RCT (ie, k=0.69; 95% CI: 0.17 to 1.21); see table 6.
ED, emergency department; RCT, randomised controlled trial; SAFE, Seizure First Aid Training For Epilepsy; TAU, treatment-as-usual.

## Safety

No serious adverse events related to participation occurred (online supplementary file 7).

## Participant feedback

Thirty-two (68.1%) patients and 20 (62.5%) SOs completed the T3 feedback questionnaire. All but one said they would participate in such a trial again; participants indicated they perceived benefits from SAFE.

## Sample size calculation

Based on the estimated effect of SAFE (see At 12 months, effect of SAFE section), table 7 shows the number of patient participants required per group for a definitive trial. If the central value in the estimate range for the dispersion parameter k of 0.69 is used and 90% power stipulated, then a total starting sample of 674 patient participants (ie, (308×2)+58) would need to be recruited. This accounts for the 9.4% of recruited patients who (on the basis of the pilot trial) would be anticipated to withdraw consent to access their routine ED data. In the pilot trial, 5 of the 53 patients recruited withdrew consent.

## Microcosting

Delivering SAFE was estimated to cost £333 (CR £288 to £380) per patient (with or without a SO). When including the cost SAFE's development (£181 per person), the mean cost per *attendee*, based on all 55 participants in SAFE+TAU or TAU groups who ultimately attended a SAFE session, was £375 (CR £348 to £402, online supplementary file 8). This reduced to £194 (CR £167 to £221) when excluding sunk costs. Staff time accounted for 50.01% of the cost of SAFE's delivery. SAFE's facilitator was not local to the trial area. Thus, an analysis of facilitator costs without travel expenses (time and cost), reduced the mean cost by 21.08% from £194 (CR £167 to £221) to £152 (CR £124 to £179) per delegate (£261 per patient with or without SO). The annual fixed cost of setting up and running SAFE was £1122 (+£35.07 per

patient, based on 32 patients/year attending with or without SO).

## DISCUSSION

The pilot trial was successful, providing estimates of key parameters, including recruitment and retention. An informed decision regarding feasibility and optimal design of a definitive trial of SAFE can now be made.

### Positives from pilot for feasibility and design of definitive trial
#### Identification, treatment and safety

The pilot trial indicates it is possible to identify, consent, randomise and safely treat persons from the target population within a definitive trial. This was not a given since no RCT had focused recruitment on them.

#### Retention and measuring ED use

By using routine data as the basis of the primary outcome measure, a definitive trial should not be affected by attrition. It permitted 12-month data to be secured for 94% of pilot participants. Despite having shorter follow-ups (6 months), 151 definitive RCTs funded by the National Institute for Health Research () secured outcome data for 89% of patients.[51] The use of routine data in our pilot trial meant it satisfied the retention progression criterion. It would not have been met if self-report data were relied on (only 67% of patients provided it). This leads us to recommend ED use be measured using routine data.

Another reason for the recommendation is its cost would be low compared with employing staff to obtain self-report data. Routine data would also prevent a definitive trial from exposure to apparent recall bias, with patients seemingly over reporting ED use (it has been asserted that bias is not an issue for "big ticket" service items).[52 53]

However, our pilot trial does caution that provision of routine data is slow and not straightforward. It took 8.5 months to obtain. In principle, such applications should be processed within 2 months.[54] Data providers should attend to such issues as those funding and designing trials need confidence that data can be secured and realistic estimates. It is also worth noting that some of the ED visits attributed to our participants may not have been epilepsy related. Since a diagnostic code is not recorded for ~35% of ED visits, we asked NHS Digital to inform us of any ED visits recorded for our participants.[25]

### Negatives from pilot trial for feasibility and design of definitive trial
#### Consent

Only a small proportion (10.6%) of identified patients were eligible and willing (12.5%) to take part, meaning the progression criterion that ≥20% of eligible patients agree to participate was not met.

The progression criterion was stated a priori because when uptake is so low there is the real possibility that those who are and are not taking part differ. We compared the age and deprivation of patients who did and did not

take part and found no obvious differences. We do not know whether they differed on other indices since access to non-participants' medical records was not ethically permissible. One indication the pilot trial might not have recruited a representative sample was the high proportion of patients seeing a neurologist in the prior 12 months (~75%). Evidence suggests this should be closer to 38%.[1]

The consent rate also raises questions over the likelihood of a definitive trial being funded since it would make it expensive. On average, each pilot site generated 17.6 patient accruals from 18 months of attendances. To achieve a sample size of ~674 PWE, a definitive trial could thus require ~39 sites (half of England's EDs). This is 2.5 times more sites than in recent NIHR funded RCTs[52] which already had a mean cost of ~£1.3 million.[55] A definitive RCT of SAFE might thus not represent acceptable value for money to funders.[56]

Epilepsy and its consequences (eg, seizures, memory difficulties, no driving license) make recruiting PWE challenging. The consent rate in the pilot is nevertheless low. In the largest RCT of self-management (Self-Management education for adults with poorly controlled epILEpsy [SMILE] trial), 37% of the people with uncontrolled epilepsy invited took part.[57] The characteristics of the patients from ED agreeing to participate in our pilot suggest that one reason for the low uptake might be stigma; 82% felt some (21.6% felt highly stigmatised). This is higher than in the wider epilepsy population; 63% of SMILE's sample felt stigmatised (12.5% highly).[58] Stigma can make PWE feel ashamed and reluctant to talk about epilepsy. This could explain why the target population is so challenging to recruit. Unfortunately, it is unclear how to revise recruitment to mitigate against this.

Evidence-based strategies were employed in the pilot to maximise recruitment,[30] and invitation materials coproduced with patients. It is generally considered preferable for a usual care provider, with whom a patient has an established relationship, to invite a person into research. Difficulties identifying the target population (see Background section) meant we had local ED clinicians do the inviting. A future trial might consider asking EDs to identify ostensibly eligible patients, but for the general practitioners of the identified patients to do the inviting. This may boost recruitment.[59]

### Effect of intervention

Our pilot trial estimated SAFE's effect to be modest (reducing ED visits from 2.1 over 12 months to 1.8). This has negative consequences for a definitive trial, not least because it requires it to have a large sample to detect the effect. Efficacy was not our pilot trial's primary focus and the estimate may be imprecise. Indeed, it might be that those who declined to participate in the trial and who appeared to differ in some important ways, might have benefited more. Previous evidence does suggest it is in the region expected.

Specifically, only one RCT—by Sajatovic et al[60]—has considered change in ED use following epilepsy self-management.[61] Conducted in the USA, it compared 'Self-MAnagement for people with epilepsy and ahistoRy of negative health evenTs' (SMART)—an 8-session group intervention—to wait list control. No significant change between groups was found in subsequent ED/hospitalisation use. For SMART, it reduced from a mean of 1.22 by 0 .44 over the 6 months following randomisation. For the controls it reduced from 2.4 by −1.26 visits.

### Judgement regarding progression to definitive trial

Thabane et al[62] provide a framework for judging whether to progress to a definitive trial. In satisfying only one progression criteria, a definitive trial based on the pilot trial's design is not feasible. We have also not identified any modifications that will make it feasible. We therefore recommend not proceeding.

### Strengths and limitationss

Strengths include the pilot trial being reported according to guidelines,[63] allocations being concealed, that patients were recruited from sites similar to those likely for a definitive trial and we included a microcosting of SAFE's delivery.

The pilot trial is not without potential weaknesses. Most importantly, despite recruited patient participants stating sufficient ED use when screened, routine data subsequently suggested ~40% did not meet the inclusion criteria. This could limit our findings' external validity and attenuate SAFE's effect. Another potential limitation is we estimated the effect of SAFE only on the proposed primary measure. We did not estimate its wider effects, including on duration of ED visits and admissions.

### CONCLUSION

A definitive trial of SAFE is not currently feasible. Research is required to determine how people from the target population can be better recruited.

**Author affiliations**
[1]Department of Health Services Research, University of Liverpool, Liverpool, UK
[2]Department of Biostatistics, University of Liverpool, Liverpool, UK
[3]Centre for Health Economics & Medicines Evaluation, Bangor University, Bangor, UK
[4]Institute of Pharmaceutical Science, King's College London, London, UK
[5]Emergency Department, Arrowe Park Hospital, Wirral University Teaching Hospital NHS Foundation Trust, Wirral, UK
[6]MacKinnon Memorial Hospital / Broadford Hospital, NHS Highland, Broadford, UK
[7]Emergency Department, Aintree University Hospital, Liverpool University Hospitals NHS Foundation Trust, Liverpool, UK
[8]School of Health and Related Research, University of Sheffield, Sheffield, UK
[9]Department of Basic and Clinical Neuroscience, Institute of Psychiatry, Psychology & Neuroscience, King's College London, London, UK
[10]Department of Molecular and Clinical Pharmacology, University of Liverpool, Liverpool, UK

**Acknowledgements** We thank the people who kindly participated in this study and the local ED sites for their support—namely, Arrowe Park Hospital (Wirral University Teaching Hospital NHS Foundation Trust) and Aintree University Hospital and the Royal Liverpool University Hospitals (Liverpool University Hospitals NHS Foundation Trust). Other persons who contributed to this work are Mrs Juliet Bransgrove (JB, epilepsy nurse specialist), Dr Duncan Appelbe (website development/management) and Ms Gail Moors (GM, Administrator). We also

acknowledge the support we received from our Steering Committee (Professor Alasdair Grey (chair), Professor Pete Bower, Dr Paul Cooper, Mr Mike Jackson, Ms Helen Coyle, Mr Mike Perry, Mrs Linda Perry, Mrs Jayne Burton and Mr Sam Burton). The study sponsor was the University of Liverpool (reference: UoL001108) and the Clinical Trials Research Centre's registration number is 12.

**Contributors** AJN and LR conceived the study and designed it together with AGM, CTS, DAH, MM and SG. SN and CTS planned and conducted the statistical analysis, EAH and DAH led with the economic analysis. AJN was the chief investigator, with MB, JM and EM leading participant identification and DS recruitment and follow-up. AJN wrote the manuscript, with revisions and approval of the final manuscript coming from all authors.

**Funding** This project was funded by the National Institute for Health Research's Health Services and Delivery Research Programme (HS&DR Programme, project number 14/19/09). The views and opinions expressed herein are those of the authors and do not necessarily reflect those of the University of Liverpool, the HS&DR programme, the NIHR, the NHS or the Department of Health and Social Care.

**Competing interests** None declared.

**Patient and public involvement** Patients and/or the public were involved in the design, or conduct, or reporting, or dissemination plans of this research. Refer to the Methods section for further details.

**Patient consent for publication** Not required.

**Provenance and peer review** Not commissioned; externally peer reviewed.

**Data availability statement** Data are available upon reasonable request. Requests for anonymised data should be submitted to the corresponding author.

**ORCID iDs**
Adam J Noble http://orcid.org/0000-0002-8070-4352
Sarah Nevitt http://orcid.org/0000-0001-9988-2709
Steve Goodacre http://orcid.org/0000-0003-0803-8444

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
