## [Reviewer comments · BMJ Open]

ARTICLE DETAILS

TITLE (PROVISIONAL)	Seizure First Aid Training For people with Epilepsy (SAFE) frequently attending emergency departments and their significant others: results of a UK multi-centre randomised controlled pilot trial
AUTHORS	Noble, Adam; Snape, Dee; Nevitt, Sarah; Holmes, EA; Morgan, Myfanwy; Tudur-Smith, Catrin; Hughes, Dyfrig; Buchanan, Mark; McVicar, Jane; MacCallum, Elizabeth; Goodacre, Steve; Ridsdale, Leone; Marson, Anthony

VERSION 1 – REVIEW

REVIEWER	Jeremy Moeller Department of Neurology Yale School of Medicine
REVIEW RETURNED	23-Nov-2019

GENERAL COMMENTS	I think this is a valuable contribution to the literature about trials of educational interventions in epilepsy. It is thorough and systematic, and would provide critically important guidance to others who would consider similar trials in the future. I think the discussion and conclusions are appropriate, and supported by the results. I do not have any recommendations for major changes. A sentence in paragraph 3 of Page 4 does not read clearly: "In the UK at least, PWE visiting EDs can though be challenging to identify for research..." It is disappointing that there is such a modest estimated benefit from SAFE regarding ED admissions. It is outside the scope of this paper to discuss why this might be (considering the pilot study was not designed to gain a precise measurement of benefit). There may be important and meaningful differences between people who can be recruited and those who cannot. One is left wondering if the people who were not recruited would have benefitted more from the intervention.
--

REVIEWER	Wilson Tam National University of Singapore, Singapore
REVIEW RETURNED	23-Dec-2019

GENERAL COMMENTS	Comment to the authors Section 2.10 Statistical Analysis Section When I first read the first paragraph, I thought the authors used sample size formula (like those in Scheaffer, R. L. et al., 2012) to compute the sample size as they mentioned the standard term like "95% confidence interval of +/-10% and a consent rate of 20% to
--

	within a 95% confidence interval of +/-4%.” Then I found out that they indeed got the number based on some simple ratio to get the number 80. I agree 80 is a reasonable number for a pilot study but I think the paragraph could be improved to make it simpler and easier to read. For example, the presentation could be improved by first stating that around 400 eligible patients could be identified per year and expected the consent rate is around 20%, so 80 could be able to recruit. Then they could proceed to explain the size is still reasonable even with 25% dropout. The second and third paragraph described the analysis. I think it is better to follow the sequence of how the results were presented. The baseline characteristics were shown in Table 4 and descriptive statistics were used (like frequency % mean sd). Table 5 was similar. Then the main analysis was using negative binomial regression to estimate the relative risk through the Intention-To-Treat (ITT) basis. First, I think the use of ITT method involves the analysis of all trial participants who were randomized, regardless of adherence to treatment protocol (e.g. dropout/withdrawal or protocol deviations). In other words, defined this way, ITT requires either no attrition or a strategy to handle missing data (see Del Re et al 2013). However, the authors mentioned no data imputation, so how they analyzed based on the number of subjects being randomized? Second, it was mentioned “SAFE’s effect on ED use, with and without adjustment for number of ED visits prior to randomisation, was estimated on an intention-to-treat basis with no data imputation, using negative binomial regression (NBR).” Please state any reasons of using negative binomial regression? Yes I think it is Ok to use such regression but better to provide reason (say why no Poisson regression?). Why did the authors used 3 level of significance i.e., 5%, 10% and 20%? It was mentioned “The proportion of correct treatment guesses was determined and Cohen’s Kappa calculated.” What was “correct treatment guess”? Should it be “Cohen’s Kappa statistic was computed”? I understand the authors wanted to see if the self-report ED visits was a reliable estimate compared with routine data but please state clear in the text. Section 2.11 Microcosting It was mentioned “The 95% central range (CR) for costs and differences were generated using 10,000 bootstrap replications.” Since the cost was computed by MS Excel, which software did the author use to generate the bootstrap sample? Also it would be useful to state clear how the authors compute the total cost and how each variable varied. Results section Section 3.1 Participant recruitment, allocation and treatment The results were in Figure 1 or Supplementary file 3? Section 3.2 Participant demographics and epilepsy characteristics The presentation of Table 4 can be improved, it would be easier to read if the authors presented the median (min, max) at the same
--	--

	row. If no missing value for the variable, please remove the row for “missing”. It is not necessary to put (n) for each variable as most were 26, 25 & 51 (just add when they were not these 3 numbers). For continuous variables, please only report mean (sd) or median (min, max). There were only 25 or 26 for each variable, I don’t think you need to have so many parameters to indicate them. For “Number of epileptic seizures in the last 12 months”, is it possible to combine into several groups say None, 1-4, 5-9, and 10 or above? Most of the groups with 0 or 1 subjects, no point to present them one by one. Section 3.3 Participant retention I think it is the major results for the pilot study? Would it be better to show the values in a graph? Section 3.4 ED use Table 5 please refer to the comment in Table 4. Section 3.6 Blinding Indeed I don’t get what the researchers needed to guess the grouping? Section 3.9 Sample size calculation Can the authors show the Keene’s formula? Also the commonly used alpha is 5% (of course 1% would also do, just not that common). Why the authors assumed 9% patients would withdrawn, was it the same % as in this pilot study? Section 3.10 Microcosting As mentioned above, a clear computation may help the readers to understand better. Del Re AC, Maisel NC, Blodgett JC, et al. Intention-to-treat analyses and missing data approaches in pharmacotherapy trials for alcohol use disorders. BMJ Open 2013;3:e003464. doi: 10.1136/bmjopen-2013-003464 Scheaffer, R. L. et al. (2012). Elementary survey sampling (7th, Intl ed.). Pacific Grove, CA: Brooks/Cole Cengage Learning.
--	---

REVIEWER	Zehuai Wen Guangdong Provincial Hospital of Chinese Medicine, Guangzhou University of Chinese Medicine, Guangzhou, China
REVIEW RETURNED	27-Dec-2019

GENERAL COMMENTS	The authors intended to observe the implementation of the Seizure First Aid Training For people with Epilepsy (SAFE) and the feasibility of using large-scale RCT to evaluate SAFE for patients with epilepsy. Indeed, this is a well-designed, clear report of the pilot trial. The author has reported the trial results in detail according to the CONSORT reporting checklist, and the conclusion is reasonable. I have no more comments.
---

REVIEWER	LM Ho
-----------------	-------

	The University of Hong Kong
REVIEW RETURNED	06-Jan-2020

GENERAL COMMENTS	This research project aims to determine the feasibility and optimal design of a RCT on seizure first aid training for people with epilepsy (SAFE). The SAFE is a 4-hour group-based self-management course, attempting to reduce emergency medical attention for seizures. The trial lasted for one year, and outcome measures were obtained at baseline, 3-month, 6-month and 12-month. It was concluded that a definitive trial of SAFE is not feasible due to difficulties in subject recruitment. It is an important research into the feasibility of an intervention for people with epilepsy, but overall eligibility rate (10.6%) and consent rate (12.5%) were both low. This poses the difficulties of recruiting enough subjects for the RCT. More importantly, there may be differences in characteristics between those who consent and not consent. Supplementary File 4 "Demographic characteristics of eligible patients who did and did not agree to participate" provides useful information on assessing the possible selection biases. So suggest to insert it in the main body, for example, as part of Table 4. It would also be easier to have a feeling of significance if p-values can be included in Table 4 as well. The retention rate does not seem to be in the tables? Is the retention rate 94.1%? (line 60, p.10) The sample size is determined based on 90% power and 1% significance, resulting in a sample of 674. A significance of 5% is usually chosen and this will require less cases. Is there any justification to choose 1% as the significance level?
---

VERSION 1 – AUTHOR RESPONSE

REVIEWER 1

REVIEWER COMMENT 1.1: “I think this is a valuable contribution to the literature about trials of educational interventions in epilepsy. It is thorough and systematic, and would provide critically important guidance to others who would consider similar trials in the future. I think the discussion and conclusions are appropriate, and supported by the results. I do not have any recommendations for major changes.”

AUTHOR RESPONSE: Thank you.

REVIEWER COMMENT 1.2: “A sentence in paragraph 3 of Page 4 does not read clearly: “In the UK at least, PWE visiting EDs can though be challenging to identify for research...””

AUTHOR RESPONSE: Thank you for highlighting this to us. The sentence has now been corrected to:

“In the UK, PWE visiting EDs can also be challenging to identify for research since most (~62%) are not being followed-up by an epilepsy specialist, primary care providers are also not always notified of

ED visits by their patients and because EDs do not always code the reason for an attendance.” (see pg. 4, para. 3)

REVIEWER COMMENT 1.3: “It is disappointing that there is such a modest estimated benefit from SAFE regarding ED admissions....There may be important and meaningful differences between people who can be recruited and those who cannot. One is left wondering if the people who were not recruited would have benefitted more from the intervention.”

We agree with the reviewer and have inserted an additional sentence within the discussion section to raise the possibility identified by the reviewer. Namely,

“Our pilot estimated SAFE’s effect to be modest (reducing ED visits from 2.1 over 12-months to 1.8). This has negative consequences for a definitive trial, not least because it requires it to have a large sample to detect the effect. Efficacy was not though our pilot’s primary focus and the estimate may be imprecise. Indeed, it might be that those who declined to participate in the trial and who appeared to differ in some important ways, might have benefitted more. Previous evidence does though suggest it is in the region expected.” (see pg. 17, para. 2)

REVIEWER 2

REVIEWER COMMENT 2.1: “Section 2.10 Statistical Analysis Section...I agree 80 is a reasonable number for a pilot study but I think the paragraph could be improved to make it simpler and easier to read. For example, the presentation could be improved by first stating that around 400 eligible patients could be identified per year and expected the consent rate is around 20%, so 80 could be able to recruit. Then they could proceed to explain the size is still reasonable even with 25% dropout.”

AUTHOR RESPONSE: We are grateful to the reviewer for pointing out this area for possible confusion. We have now amended the section as per their instruction, namely:

“Based on existing data, it was anticipated 12 months of attendances from each ED would allow identification of ~400 eligible patients. With a 20% consent rate, 80 patient participant accruals could be expected. With 80 patient participants we could estimate a potential drop-out rate of 25% to within a 95% confidence interval of +/-10% and a consent rate of 20% to within a 95% confidence interval of +/-4%. Assuming ED data at T3 was not available for 25% of patients, data from 60 patients would still allow robust estimation of the ED rate and dispersion parameter. Sample sizes of 24 to 50 have been recommended as ‘adequate’ for pilot studies.” (see pg. 8, para. 3)

REVIEWER COMMENT 2.2: “The second and third paragraph described the analysis. I think it is better to follow the sequence of how the results were presented. The baseline

characteristics were shown in Table 4 and descriptive statistics were used (like frequency % mean sd). Table 5 was similar.

AUTHOR RESPONSE: As per the reviewer's comment we have slightly amended the order in which the elements of analysis are described so as to correspond with the order in which the results are presented. (please see section 2.10, pg. 8-10).

REVIEWER COMMENT 2.3: "...the main analysis was using negative binomial regression to estimate the relative risk through the Intention-To-Treat (ITT) basis. First, I think the use of ITT method involves the analysis of all trial participants who were randomized, regardless of adherence to treatment protocol (e.g. dropout/withdrawal or protocol deviations). In other words, defined this way, ITT requires either no attrition or a strategy to handle missing data (see Del Re et al 2013). However, the authors mentioned no data imputation, so how they analyzed based on the number of subjects being randomized?"

AUTHOR RESPONSE: Thank you to the reviewer for pointing out the multiple definitions of ITT analysis (Del Re et al 2013). In this context, no exclusions were made based on adherence or treatment protocol, but we did not impute data for any participants where the number of ED visits was missing. This approach actually meets the definition of a modified ITT analysis. In light of the reviewer's comment we now more clearly state this within our manuscript. Specifically:

"SAFE's effect on ED use, with and without adjustment for number of ED visits prior to randomisation, was estimated on a modified intention-to-treat basis, including participants with their number of ED visits recorded, with no data imputation, using negative binomial regression (NBR)." (see pg. 9, para. 2)

REVIEWER COMMENT 2.4: "Please state any reasons of using negative binomial regression? Yes I think it is Ok to use such regression but better to provide reason (say why no Poisson regression?)."

AUTHOR RESPONSE: We have added the following sentence:

"NBR was the pre-specified statistical approach as over-dispersion (i.e. large variance) was anticipated in the number of ED visits reported." (see pg. 9, para. 2)

REVIEWER COMMENT 2.5: "Why did the authors used 3 level of significance i.e., 5%, 10% and 20%?"

AUTHOR RESPONSE: When completing one of the project's secondary objectives of reporting the estimated effect of SAFE on the primary outcome measure, we reported the statistical significance of this effect at the 5, 10 and 20% alpha level. The reason for reporting at 3 levels of significance is that pilot studies are typically underpowered to allow definitive hypothesis testing at the 5% alpha

level. Despite this, there can be a tendency to focus on the results of hypothesis testing undertaken by a pilot trial. It is recommended by some therefore that less stringent alpha levels should be considered to allow for a more comprehensive evaluation of the potential effect of the intervention, to encourage greater focus on wider parameters, such as the meaningfulness of the effect and to avoid not progressing to a full trial when in fact the effect size would suggest otherwise.

In view of the reviewer's comment we have now made the issue clearer and provide a reference for the reader who requires more technical information on the matter::

"Between-group differences are presented as Rate Ratios and, as per recommendations for hypothesis testing within pilot trials,⁴² tested according to 5, 10 and 20% levels of significance." (see pg. 9, para. 2)

REVIEWER COMMENT 2.6: "It was mentioned "The proportion of correct treatment guesses was determined and Cohen's Kappa calculated." What was "correct treatment guess"? Should it be "Cohen's Kappa statistic was computed"?"

AUTHOR RESPONSE: We have amended section 2.9 to incorporate the changes the reviewer proposes and also additional changes to bring clarity to the issue. We now note:

"To evaluate blinding, DS completed a 'Treatment Guess' form after each participant's T3 assessment or withdrawal. It required her to state which treatment arm she believed the participant had been randomised to." (see pg. 8, para. 2)

REVIEWER COMMENT 2.7: "I understand the authors wanted to see if the self-report ED visits was a reliable estimate compared with routine data but please state clear in the text."

AUTHOR RESPONSE: In light of the reviewer's comment, we now note the following within Section 2.9:

"To see if self-reported ED visits provided a reliable estimate compared to routine data the agreement between the two data sources on how many ED visits a patient had made was explored." (see pg. 8, para. 2)

REVIEWER COMMENT 2.8: "Section 2.11...It was mentioned "The 95% central range (CR) for costs and differences were generated using 10,000 bootstrap replications." Since the cost was computed by MS Excel, which software did the author use to generate the bootstrap sample? Also it would be useful to state clear how the authors compute the total cost and how each variable varied."

AUTHOR RESPONSE: The 95% central range (CR) for costs and differences were generated using 10,000 Monte Carlo Simulations, in MS Excel. Full details of each cost, disaggregated to an

individual group level, are provided in Supplementary File 8. We have also revised the narrative in section 2.11 to provide further details. (see pg. 10, para. 2)

REVIEWER COMMENT 2.9: “Section 3.1 Participant recruitment, allocation and treatment The results were in Figure 1 or Supplementary file 3?”

AUTHOR RESPONSE: Both Figure 1 and Supplementary File 3 (Supplementary File 2 in the revised manuscript) present information pertinent to this point. Figure 1 presents the headline information, whilst Supplementary File 3 a more detailed breakdown relating to reasons for exclusions at the different recruitment stages. We have amended the text slightly to make it clear that both are relevant and that reference to the two is not a typographic error. Specifically, we now note: “Of the 4016 individuals identified, 1220 (30.4%, CI 29.0% to 31.8%) had visited for established epilepsy. Of these, 424 (34.8%, CI 32.1% to 37.4%) were eligible; eligibility rate 10.6% (CI 9.6% to 11.5%) (Figure 1 and Supplementary File 2).” (see pg. 11, para. 2)

REVIEWER COMMENT 2.10: “Section 3.2 Participant demographics and epilepsy characteristics. The presentation of Table 4 can be improved, it would be easier to read if the authors presented the median (min, max) at the same row. If no missing value for the variable, please remove the row for “missing”. It is not necessary to put (n) for each variable as most were 26, 25 & 51 (just add when they were not these 3 numbers). For continuous variables, please only report mean (sd) or median (min, max). There were only 25 or 26 for each variable, I don’t think you need to have so many parameters to indicate them. For “Number of epileptic seizures in the last 12 months”, is it possible to combine into several groups say None, 1-4, 5-9, and 10 or above? Most of the groups with 0 or 1 subjects, no point to present them one by one.

AUTHOR RESPONSE: We thank the reviewer for their suggestion. As per their suggestion we now present the median (min, max) on the same row, have removed the missing value row when this is not relevant, removed the group numbers on each row and inserted this only at the top of the column and grouped seizures in the last year by a category (namely, 0-3, 4-6, 7-9, 10 or more). We have chosen these subcategories as it has been used in prior trials of self-management for epilepsy (e.g., Ridsdale et al. *Epilepsia*. 2018 May;59(5):1048-1061) and so permits direct comparison of the samples within recent trials.

We have not though only presented the mean or median for the continuous variables as requested by the reviewer. We have chosen to continue to present both since the nature of some of the variables was quasi-continuous and also because the difference between the mean and median is important for the reader to see to have a clearer understanding of the distribution.

(Please note that these changes have not been tracked due to limitations with this function when working with tables)

REVIEWER COMMENT 2.11: "Section 3.3 Participant retention.. I think it is the major results for the pilot study? Would it be better to show the values in a graph?"

AUTHOR RESPONSE: It is certainly one of the major results. However, with respect, we are not convinced that a graph and the space it would occupy is warranted given the result is a single number (i.e. a proportion).

REVIEWER COMMENT 2.12: "Section 3.4 ED use...Table 5 please refer to the comment in Table 4."

AUTHOR RESPONSE: We have made the recommended changes to Table 5.

REVIEWER COMMENT 2.13: "Section 3.6 Blinding... I don't get what the researchers needed to guess the grouping?"

AUTHOR RESPONSE: The researcher was, as per the reviewer's comment, required to guess to which treatment arm the patient participant had been randomised to. We have now clarified this. We have also made further relevant changes in section 2.9 following comment 2.6 from the reviewer. We now state:

"The researcher correctly guessed which of the two treatment arms the allocation of 35 patient participants had been allocated to by the randomisation process; unblinding rate 68.6% (CI 54.1% to 80.9%). The chance-corrected kappa statistic of 0.37 (CI 0.12 to 0.63) equated to "fair" agreement." (see pg. 13, para. 4)

REVIEWER COMMENT 2.14: "Section 3.9 Sample size calculation...Can the authors show the Keene's formula? Also the commonly used alpha is 5% (of course 1% would also do, just not that common). Why the authors assumed 9% patients would withdrawn, was it the same % as in this pilot study?"

AUTHOR RESPONSE: Within Section 2.10 we now provide the formula (see pgs. 9-10) and in Section 3.9 the workings and output of the calculations (see pg. 14, para. 1). We note that an alpha level of 5% was in fact used for the sample size calculations, but we incorrectly reported this at 1%. We have now corrected this (see Section 2.10 and 3.9) and are grateful to the reviewer for highlighting it.

We can also confirm that the 9% withdrawal rate is indeed taken from the pilot trial. We have now made this clearer within Section 3.9 where we now state:

*“Based on the estimated effect of SAFE (see 3.4.2), Table 7 shows the number of patient participants required per group for a definitive trial. If the central value in the estimate range for the dispersion parameter k of 0.69 is used and 90% power stipulated, then a total starting sample of 674 patient participants (i.e., $[308*2]+58$) would need to be recruited. This accounts for the 9.4% of recruited patients who (on the basis of the pilot trial) would be anticipated to withdraw consent to access their routine ED data. In the pilot, 5 of the 53 patients recruited withdraw consent.”* (see pg. 14, para. 1)

REVIEWER COMMENT 2.15: “Section 3.10 Microcosting...As mentioned above, a clear computation may help the readers to understand better.”

AUTHOR RESPONSE: Many thanks for your suggestion. The narrative in section 2.11 of the manuscript has been revised to include further details. (see pg. 10, para. 2)

REVIEWER 3

REVIEWER COMMENT 3.1: “...this is a well-designed, clear report of the pilot trial. The author has reported the trial results in detail according to the CONSORT reporting checklist, and the conclusion is reasonable. I have no more comments.”

AUTHOR RESPONSE: Thank you.

REVIEWER 4

REVIEWER COMMENT 4.1: “...Supplementary File 4 “Demographic characteristics of eligible patients who did and did not agree to participate” provides useful information on assessing the possible selection biases. So suggest to insert it in the main body, for example, as part of Table 4.

AUTHOR RESPONSE: As per the reviewer’s suggestion, Supplementary File 4 has been integrated into Table 4.

REVIEWER COMMENT 4.2: “It would also be easier to have a feeling of significance if p-values can be included in Table 4 as well.”

AUTHOR RESPONSE: We thank the reviewer for this suggestion, but we have followed the guidance that statistical testing of baseline characteristics of randomised groups is not recommended as such tests really only assess the success of randomisation rather than the effect of any observed imbalance of characteristics on the results of the trial (e.g., Altman, DG. "Comparability of randomised groups." *Journal of the Royal Statistical Society: Series D The Statistician* 34.1, 1985: 125-136.). Instead, we have commented on any observed imbalanced in characteristics (Section 3.2).

REVIEWER COMMENT 4.3: “The retention rate does not seem to be in the tables? Is the retention rate 94.1%? (line 60, p.10)”

AUTHOR RESPONSE: The retention rate is indeed 94.1% when operationalised as a patient participant having ED data at 12 months based on routine data. To make this clearer we have now amended the wording in 3.3.1 and 3.3.2. In 3.3.1, we now, for example, state:

“3.3 Participant retention

3.3.1 Proposed primary outcome measure:

Of the randomised patients, 3 (5.8%) formally withdrew over follow-up, removing access to their routine data. Primary outcome data on ED use at 12 months (and for the 12 months prior to randomisation) was available for the remaining 48 patients, giving a retention rate of 94.1%.

3.3.2 Proposed secondary outcome measures:

Thirty-seven (72.5%) randomised patients and 21 (56.8%) SOs attended their 12-months questionnaire appointment (T3). The extent to which measures were completed at these and the interim appointments varied (Supplementary File 5). Self-report data on ED use at T3 was obtained from only 34 patients, giving a retention rate on this measure of 66.7% patients.” (see pg. 12, paras. 3-4)

REVIEWER COMMENT 4.4: “The sample size is determined based on 90% power and 1% significance, resulting in a sample of 674. A significance of 5% is usually chosen and this will require less cases. Is there any justification to choose 1% as the significance level?”

AUTHOR RESPONSE: We are grateful to the reviewer for raising this. We did in fact base the sample size calculations on the use of an alpha level of 5%, but incorrectly reported the 1% level was used. We have now corrected this and thank for the reviewer for highlighting it. We thus now note:

*“Based on the estimated effect of SAFE (see 3.4.2), Table 7 shows the number of patient participants required per group for a definitive trial. If the central value in the estimate range for the dispersion parameter k of 0.69 is used and 90% power stipulated, then a total starting sample of 674 patient participants (i.e., $[308*2]+58$) would need to be recruited. This accounts for the 9.4% of recruited patients who (on the basis of the pilot trial) would be anticipated to withdraw consent to access their routine ED data.” (see pg. 14, para. 2)*

FORMATTING AMENDMENTS

AMENDMENT REQUEST 1: “The author “T Marson” in your main document is registered as “Marson, Anthony Guy” in ScholarOne. Please ensure that the author has same registered name.”

AUTHOR RESPONSE: This has now been made consistent, with T Marson, now being changed within the manuscript to AG Marson.

AMENDMENT REQUEST 2: “Please re-upload your supplementary files in PDF format.”

AUTHOR RESPONSE: Done.

AMENDMENT REQUEST 3: “Supplementary file citations should be in ascending order.”

AUTHOR RESPONSE: The file citations were in ascending order when the footnote to Table 2 is considered (we there made reference to a supplementary file). However, we appreciate one cannot guarantee where Table 2 will appear when published. We have thus correct the file citation order in line with your instructions.

VERSION 2 – REVIEW

REVIEWER	Wilson Tam National University of Singapore, Singapore
REVIEW RETURNED	18-Feb-2020

GENERAL COMMENTS	The authors have addressed all my previous comments and I have no further comments.
---

REVIEWER	LM Ho The University of Hong Kong
REVIEW RETURNED	14-Feb-2020

GENERAL COMMENTS	1. The sample size estimation was revised and significantly improved. Sufficient details were also given in the Microcosting section. 2. Although presenting mean, SD, median etc may help readers to have a clearer understanding of the distribution, the tables seems over crowded. As the sample size in each arm is around 25, the shape of the distribution is highly affected by sampling errors, and thus having too much detail may mislead readers indeed. If all statistics are considered to be useful, it may be better to use IQR instead of min and max for which are easily influenced by outliers. 3. There was considerable proportion of subjects declining to participate in the trial, which may lead to biases. Other than their reasons for declining participation, is there any way to compare their demographics, eg age, sex education, and employment? This can be useful to better understand them, and assess the possible biases. 4. What is the meaning of a “modified intention-to-treat basis” in the Statistical Analysis section. Please explain.
---

VERSION 2 – AUTHOR RESPONSE

REVIEWER 4

REVIEWER COMMENT 4.1.: “The sample size estimation was revised and significantly improved. Sufficient details were also given in the Microcosting section.”

AUTHOR RESPONSE:

Thank you for noting this.

REVIEWER COMMENT 4.2.: “Although presenting mean, SD, median etc may help readers to have a clearer understanding of the distribution, the tables seems over crowded. As the sample size in each arm is around 25, the shape of the distribution is highly affected by sampling errors, and thus having too much detail may mislead readers indeed. If all statistics are considered to be useful, it may be better to use IQR instead of min and max for which are easily influenced by outliers.”

AUTHOR RESPONSE:

It is our position that showing the minimum and maximum values is helpful as it shows how variable the sample was in some instances and provides a high degree of transparency. The same goes for the other statistics included in the table and our intention report the data in this way in our pre-specified statistical analysis plan (SAP can be made available on request). For this reason, and given that none of the other 3 reviewers of the manuscript had concerns about the style of table presentation, we politely decline the reviewer’s invitation to modify them.

REVIEWER COMMENT 4.3.: “There was considerable proportion of subjects declining to participate in the trial, which may lead to biases. Other than their reasons for declining participation, is there any way to compare their demographics, eg age, sex education, and employment? This can be useful to better understand them, and assess the possible biases.”

AUTHOR RESPONSE:

The analysis proposed is indeed important. It is though something we already included in our manuscript. Specifically, we presented data for visual inspection that allows for the sex, age at presentation to emergency department (years) and deprivation (Index of Multiple Deprivation, IMD) of those who did and did not agree to participate in the trial to be compared. We note that we did this in the third paragraph of Section 2.10 of our article and present the findings in Section 3.4 and Table 4. In case it is of interest, in our manuscript we noted that: “Recruited patients were comparable in age and deprivation to those declining participation. Females were though over-represented (Table 4)”.

REVIEWER COMMENT 4.4.: “What is the meaning of a “modified intention-to-treat basis” in the Statistical Analysis section. Please explain.”

AUTHOR RESPONSE:

We included this statement in the revised version of the manuscript in response to a comment from Reviewer 2 (who on the basis of them not making any further comments appears satisfied with our revision). Specifically, Reviewer 2 highlighted that there are a number of definitions as to what comprises an ‘intention-to-treat’ analysis and highlighted work on this issue by Del Re et al. (BMJ open. 2013 Nov 1;3(11):e003464). In light of Reviewer 2’s comment we therefore took the opportunity in our revision to offer greater clarity as to what specific ITT approach we used. In our pilot trial analysis, no exclusions were made based on adherence or treatment protocol, but we did not impute data for any participants where the number of ED visits was missing. Such an approach is typically defined as constituting a modified, rather traditional ITT analysis. In light of Reviewer 4’s new

comment, to bring further clarify to the issue we have now slightly revised the manuscript text further and referring the interested reader to Del Re et al.'s work. Specifically, in Section 2.10 we now note:

“SAFE’s effect on ED use, with and without adjustment for number of ED visits prior to randomisation, was estimated using negative binomial regression (NBR) on a modified intention-to-treat basis (as defined by Del Re et al.⁴²). Participants were included with their number of ED visits recorded with no data imputation. NBR was the pre-specified statistical approach as over-dispersion (i.e. large variance) was anticipated in the number of ED visits reported. Between-group differences are presented as Rate Ratios and, as per recommendations for hypothesis testing within pilot trials,⁴³ tested according to 5, 10 and 20% levels of significance.”